# Transcriptional Regulation and Gene Mapping of Internode Elongation and Late Budding in the Chinese Cabbage Mutant *lcc*

**DOI:** 10.3390/plants13081083

**Published:** 2024-04-12

**Authors:** Yunqin Zhang, Shuxin Xuan, Jiaojiao Zhao, Hui Li, Yin Lu, Rui Li, Yanhua Wang, Shuxing Shen, Xiaoxue Sun, Daling Feng

**Affiliations:** 1State Key Laboratory of North China Crop Improvement and Regulation, Key Laboratory of Vegetable Germplasm Innovation and Utilization of Hebei, College of Life Science, Hebei Agricultural University, Baoding 071000, China; zhangyunqinpaper@163.com (Y.Z.); lr990901@163.com (R.L.); 2Key Laboratory of Vegetable Germplasm Innovation and Utilization of Hebei, College of Horticulture, Hebei Agricultural University, Baoding 071000, China; yyxsx@hebau.edu.cn (S.X.); zhaojiaojiaott@163.com (J.Z.); haley9@foxmail.com (H.L.); luagzoujidong@163.com (Y.L.); yywyh@hebau.edu.cn (Y.W.); shensx@hebau.edu.cn (S.S.); 3State Key Laboratory of North China Crop Improvement and Regulation, Key Laboratory of Vegetable Germplasm Innovation and Utilization of Hebei, College of Horticulture, Hebei Agricultural University, Baoding 071000, China

**Keywords:** Chinese cabbage, internodes, budding time, BSA, QTL

## Abstract

Two important traits of Chinese cabbage, internode length and budding time, destroy the maintenance of rosette leaves in the vegetative growth stage and affect flowering in the reproductive growth stage. Internodes have received much attention and research in rice due to their effect on lodging resistance, but they are rarely studied in Chinese cabbage. In Chinese cabbage, internode elongation affects not only the maintenance of rosette leaves but also bolting and yield. Budding is also an important characteristic of Chinese cabbage entering reproductive growth. Although many studies have reported on flowering and bolting, studies on bud emergence and the timing of budding are scarce. In this study, the mutant *lcc* induced by EMS (Ethyl Methane Sulfonate) was used to study internode elongation in the seedling stage and late budding in the budding stage. By comparing the gene expression patterns of mutant *lcc* and wild-type A03, 2280 differentially expressed genes were identified in the seedling stage, 714 differentially expressed genes were identified in the early budding stage, and 1052 differentially expressed genes were identified in the budding stage. Here, the transcript expression patterns of genes in the plant hormone signaling and clock rhythm pathways were investigated in relation to the regulation of internode elongation and budding in Chinese cabbage. In addition, an F_2_ population was constructed with the mutants *lcc* and R500. A high-density genetic map with 1602 marker loci was created, and QTLs for internode length and budding time were identified. Specifically, five QTLs for internode length and five QTLs for budding time were obtained. According to transcriptome data analysis, the internode length candidate gene *BraA02g005840.3C* (*PIN8*) and budding time candidate genes *BraA02g003870.3C* (*HY5-1*) and *BraA02g005190.3C* (*CHS-1*) were identified. These findings provide insight into the regulation of internode length and budding time in Chinese cabbage.

## 1. Introduction

Chinese cabbage (*Brassica rapa* ssp. *pekinensis*) is an important member of the Brassicaceae family and is popular worldwide, with a long history of cultivation [1]. The growth and development of Chinese cabbage include the vegetative stages (seedling stage, rosette stage, and heading stage) and reproductive stages (budding stage, bolting stage, and flowering stage). In the vegetative stages, internodes are not visible in Chinese cabbage. An internode is the section of stem between two nodes. Internodes are crucial for plants because all plants have a stem organ, which supports the whole body of the plant and possesses complicated internal organizational structure, functioning as a pipe transporting water and nutrients [2]. Internodes can also change plant architecture [3]. For many crops and vegetables, plants with long internodes and tall stems are easily knocked down by rain or wind, which indirectly affects yield [4].

In Chinese cabbage, the elongation of internodes in the vegetative stage seriously affects its commercial properties and leafy head formation, which limit production, reduce economic benefits, and damage morphological construction and plant height. However, there are few studies on internode development in Chinese cabbage. Internode elongation could result from cell proliferation and cell extension, which are controlled by both hormone and genetic factors [5,6]. There have been some studies on internode development in other crops, such as rice. In deepwater rice, the internodes elongate via the participation of gibberellic acid (GA) biosynthesis and signal transduction [7]. Overexpression of *OsbHLH073* in rice resulted in shortened internodes and shorter plant height by modulating GA homeostasis [8]. In rice, the *osarp6* mutant showed a shorter internode length and dwarfism due to a reduced number of internode cells [9]. OsBRI1 promoted the formation of the intercalary meristem and the longitudinal elongation of internode cells in rice [10]. Meanwhile, *ath1* mutant Arabidopsis seedlings with longer internodes showed that *Ath1* mediated the PIF pathway by directly activating BOP1 and BOP2, whose products destabilize PIF proteins to affect internodes [11]. Internodes are associated with the shade avoidance response (SAR), which is triggered by an increase in far-red (FR) light reflected by neighboring plants [12]. MiR172-mediated restriction of AP2 may modulate the jasmonate pathway to facilitate gibberellin-promoted stem growth during flowering [13]. Internode elongation in rice and Arabidopsis is closely associated with plant hormone regulation. However, little is known about the relationship between internode elongation and plant hormone regulation in Arabidopsis, let alone Chinese cabbage.

Another important trait of Chinese cabbage is budding. The budding time affects the formation of leafy heads and affects the flowering time of Chinese cabbage. Flowering time is impacted by genes in many pathways and environmental cues, such as the classical photoperiod pathway, vernalization pathway, autonomous pathway, GA pathway, and newly identified age pathway [14]. The formation of a flower in Arabidopsis requires many stages, and the phase before the first bud opens is called the budding period; flowers always emerge on the flanks of the shoot apical meristem (SAM) [15]. Budding represents the transition of the apical meristem from vegetative to reproductive growth. The budding period will shorten the heading stage of Chinese cabbage, which will cause the leafy head to be loose or even prevent head formation. Although many studies have been performed on flowering time, few studies have been performed on budding in Chinese cabbage.

In this study, we identified a Chinese cabbage EMS mutant, *lcc*, which has a long internode length and later budding time than wild-type A03. Using *lcc* as the material, an analysis of the genome and transcriptome was performed, which revealed the regulatory pathways regulating internode development during the vegetative stage and budding during the reproductive stage and allowed the identification of related candidate genes. The analysis of these two traits, which have historically received less attention than other traits, was performed to carry out research on Chinese cabbage from a new perspective.

## 2. Results

### 2.1. Transcriptome Analysis Reveals Genetic Factors Underlying Internode Length and Budding Time Regulation in Chinese Cabbage

After mutagenesis, the internode length of *lcc* was longer than that of A03, while the budding time was a week later than that of A03 (Figure 1a,b). Differential expression (DE) analysis was performed between *lcc* and A03 at three different stages: the seedling stage (T1), early budding stage (T2), and budding stage (T3). There were 2280 differentially expressed genes (DEGs) at the seedling stage (1599 upregulated and 681 downregulated), 714 DEGs at the early budding stage (364 upregulated and 350 downregulated), and 1052 DEGs at the budding stage (245 upregulated and 807 downregulated). Forty-one DEGs were shared among the three time points (Figure 1c). The seedling stage had the most DEGs, and the early budding stage had the fewest DEGs (Figure 1c).

#### 2.1.1. Plant Hormone Gene Transcripts Modulated Internode Growth

At the seedling stage, we aimed to assess the DEGs to determine those related to internode length. Among the top 10 enriched Kyoto Encyclopedia of Genes and Genomes (KEGG) pathways of DEGs, hormone regulatory pathways were associated with the largest number of DEGs (49), which included various regulatory factors and hormone response genes (Figure 1d, Appendix A).

Plant hormone signaling networks play different roles in plant growth. Cytokinin (CK) was related to cell division; jasmonic acid (JA), GA, and brassinosteroid (BR) affected cell wall remodeling; and auxin (IAA) was related to cell elongation (Figure 1e). We found that in the CK signal transduction pathway, *PHOSPHOTRANSMITTER 4* (*AHP4*) was upregulated, and *RESPONSE REGULATOR 16* (*ARR16*) was downregulated. In the JA signaling pathway, two *JASMONATE RESISTANT 1* (*JAR1-1*, *JAR1-2*) and two *JASMONATE ZIM-Domain* (*JAZ3*, *JAZ6*) genes were upregulated, and one *JAZ* (*JAZ10*) gene was downregulated. At the seedling stage, one gene, *GA INSENSITIVE DWARF1B* (*GID1B*), related to GA signaling, was upregulated. In BR signaling, one *BRI1-ASSOCIATED RECEPTOR KINASE* (*BAK1*) gene, one *BRASSINAZOLE-RESISTANT 2* (*BZR2*) gene and three *XYLOGLUCAN ENDOTRANSGLUCOSYLASE/HYDROLASE 24* genes (*XTH24-1*, *XTH24-2*, *XTH24-3*, and *XTH24*, also named *TCH4*, which encodes enzymes that elongate cells by modifying the cell wall) were upregulated at the seedling stage. In IAA signaling during the seedling stage, one *indole-3-acetic acid inducible 14* (*IAA14*) gene and ten *SMALL AUXIN UPREGULATED RNA* (*SAUR10*, *SAUR19*, *SAUR20-1*, *SAUR20-2*, *SAUR20-3*, *SAUR21-1*, *SAUR21-2*, *SAUR21-3*, *SAUR22*, and *SAUR30*) genes were upregulated, and one *AUXIN1* gene, one *MALE-GAMETE-SPECIFIC HISTONE H3* (*GH3.6*) gene, and two *AUXIN RESPPONSE FACTOR* (*ARF5-1*, *ARF5-2*) genes were downregulated. Some genes affecting plant cell division, elongation, and expansion could be regulated by hormone signal transduction pathways, such as *EXPs* and *XTHs*. Two *EXP* (*EXPA6*, *EXP8*) genes and *XTH9* were also differentially expressed during the seedling stage but were not involved in any hormone signal transduction. The plant hormone signal transduction pathway also cooperates with other genes to regulate plant cell development, such as *PHYTOCHROME INTERACTING FACTOR* (*PIF8*), *BOTRYTIS SUSCEPTIBLE1 INTERACTOR* (*BOI1*), and *SCARECROW-LIKE 3* (*SCL3*), as shown in Figure 1e.

#### 2.1.2. Expression Analysis of Genes Influencing Budding Time in Chinese Cabbage

At the early budding and budding stages, the DEGs between A03 and the mutant *lcc* were enriched in the clock rhythm pathway (KEGG map04712). Interestingly, the DEGs were upregulated at the early budding stage but downregulated at the budding stage (Figure 2a,b, Appendix A). At the early budding stage, there were 10 DEGs associated with the circadian clock, including *PSEUDO-RESPONSE REGULATOR 9* (*PRR9-1*, *PRR9-2*), *CRYPTOCHROME 2* (two *CRY2s* in Chinese cabbage, where *BraA10g002940.3C* was differentially expressed and *BraA08g034900.3C* was not), *CIRCADIAN CLOCK ASSOCIATED 1* (*CCA1*), *CHALCONE SYNTHASE* (*CHS-1*, *CHS-2*, *CHS-3*), *SUPPRESSOR OF PHYA-105 1* (*SPA1*, two *SAP1s* in Chinese cabbage, where *BraA03g024140.3C* was differentially expressed and *BraA05g001450.3C* was not), *CYCLING DOF FACTOR 1* (two *CDF1s* in Chinese cabbage, where *BraA02g043680.3C* was differentially expressed and *BraA03g043720.3C* was not), and *ELONGATED HYPOCOTYL 5* (*HY5-2*, *BraA05g029990.3C*). At the budding stage, the DEGs related to the circadian clock were *PRR9-2*, one *TOC1* (*PRR1*, *PSEUDO-RESPONSE REGULATOR 1*, *BraA03g044360.3C*, two *TOC1s* in Chinese cabbage), *GIGANTEA* (*GI*), *CHS-1*, one *CONSTITUTIVE PHOTOMORPHOGENIC 1* (*COP1*, *BraA09g006460.3C*, two *COP1s* in Chinese cabbage), *TWIN SISTER OF FT* (*TSF*), and *HY5* (*HY5-1*, *HY5-2*), all of which were downregulated. In contrast, only PRR9-2 (*BraA05g001070.3C*), *HY5-2* (*BraA05g029990.3C*), and *CHS-1* (*BraA02g005190.3C*), which are involved in the regulation of auxin transport, were shared by the early budding stage and budding stage. Numerous studies on flowering have reported that flowering is controlled and regulated by photoperiod, GA, vernalization, temperature, age, and autonomy (Figure 2a,b). The expression of *GID1b* and *TSF* showed the same trend (upregulated in T1, downregulated in T3), whereas *GID1a* was differentially expressed only in T2, and the circadian clock gated the oscillation of *GID1*. In the early budding stage, *GID1a* and *SOC1* were upregulated, and *GA20OX3-1* was downregulated. In the budding stage, *GID1b* and *GA20X6* were downregulated, and *GA20OX3-2* and *FLC* were upregulated.

For further analysis of budding time (BT), we identified groups of genes that shared similar transcriptional profiles across the three time points. Clustering of all DEGs resulted in ten subsets according to the expression trend in A03 and *lcc* (Figure 3a,b). We found that the circadian clock-related genes mentioned above were scattered among clusters 1, 2, 6, 8, 9, and 10; flowering-related genes, such as *SOC1*, *TSF*, and *FLC*, were in clusters 2, 8, 9, and 10. DEGs in clusters 2, 8, and 10 might be coregulated BT. In addition, we checked the expression of *BraA02g005190.3C* (*CHS-1*), *BraA10g024990.3C* (*CHS-3*), *BraA03g044360.3C* (*TOC1*), *BraA07g031650.3C* (*TSF*), *BraA02g003870.3C* (*HY5-1*), *BraA05g029990.3C* (*HY5-2*), and *BraA05g001070.3C* (*PRR9-2*) using RT–qPCR to verify the accuracy of the transcriptome data (Figure 3c, Table 1). The expression of these genes was not different from the results obtained with RNA-Seq, which indicated that the RNA-Seq data were reliable.

### 2.2. QTL Analysis of Internode Length and Budding Time Traits

#### 2.2.1. Phenotyping Data Analysis of Parental Lines and the F_2_ Population

*Lcc* and R500 showed different hypocotyl, internode, leaf hair, and budding time phenotypes. *Lcc* was characterized by long hypocotyls, short internodes, and no budding and no flowering without vernalization. R500 is an oil-type Chinese cabbage whose hypocotyls were shorter than those of *lcc* according to a t test (Figure 4a, *p* < 0.05, Table 2). The internodes of R500 were longer than those of *lcc* (Figure 4b), and the budding time and flowering time of R500 were earlier than those of *lcc*; detailed statistics are shown in Appendix A. Compared with the parental lines, the F_1_ hybrids had intermediate hypocotyls and internodes but displayed budding and flowering with vernalization. The hypocotyl length of F_2_ was normally distributed (Appendix A), and there were three types of internodes in F_2_: long internodes such as those in R500, internodes similar to those of the F_1_ generation, and short internodes such as those of *lcc* (Figure 4b). The budding time (BT) of F_2_ ranged from 1 to 90 and is shown in Appendix A.

#### 2.2.2. A High-Density Genetic Linkage Map for the F_2_ Population

We sequenced *lcc*, R500, and 150 of their F_2_ individuals and obtained 376,474 polymorphic markers (“AA × BB” type). After filtering, a total of 5597 valid markers were retained and used for linkage analysis using Joinmap 4. A high-density genetic map of 10 linkage groups (A01–A10) formed by 1602 markers was obtained for further study (Appendix A). The total length of the genetic map was 1260.271 centimorgans (cM), with an average intermarker distance of 0.79 cM (Appendix A). A03 was the longest chromosome, at 174.437 cM, with an average marker spacing of 0.72 cM. A08 was the shortest chromosome, at 92.235 cM, and the average marker spacing was 0.63. Although the sequence of SNP markers in the linkage group and the Chiifu physical map of the reference genome were similar, chromosome translocations existed in A01 and A05 (Figure 4c).

#### 2.2.3. QTL Mapping

We detected QTLs for hypocotyl length (HL), internode length (IN), leaf hairs (LH), and BT in the 10 linkage groups by applying the IM method and MQM method (Table 1).

Thirteen IM-QTLs were found using the IM method (Table 1, Figure 4d). For HL, two QTLs were identified on A02 and A07. The logarithm of the odds (LOD) scores were 3.35 and 3.48, respectively. The explained variation (Exp) per QTL was 9.9% and 10.3%. Five QTLs for IN were identified, including IN-IM1 on A01 (3.06), IN-IM2, and IN-IM3 on A02 (4.29 and 4.38), IN-IM4 on A05 (3.71) and IN-IM5 on A07 (3.39). The Exp for IN-IM ranged from 9.2–11%, and the total amount of variation explained was 55.9%. For LH, one QTL was located on A06 with an LOD score of 23.63 and an Exp of 52.5%. For BT, five QTLs were surveyed on A02 (BT-IM1), A03 (BT-IM2), A06 (BT-IM3), A09 (BT-IM4), and A10 (BT-IM5). The LOD score ranged from 3.2 to 7.58, and the phenotypic variance explained using these QTLs ranged from 17.4% to 37.6%.

Seven MQM-QTLs for HL, IN, LF, and BT were identified by performing the MQM method (Table 1, Figure 4d). One QTL linked to HL located on A07 was identified with an LOD score of 3.01 and an Exp of 8.1%. For IN, four QTLs were identified: IN-MQM1 on A01, IN-MQM2, and IN-MQM3 on A02, and IN-MQM4 on A05. The LOD scores ranged from 3.19 to 4.38, and the Exp ranged from 7% to 12.9%. One QTL associated with LH was found on A06 with an LOD score of 19.12 and an Exp of 35.3%. One QTL for BT was identified on A02; the LOD score was 4.89, and the Exp was 15.8%.

#### 2.2.4. Differentially Expressed Genes within QTLs

QTLs for the target phenotypes (IN and BT) overlapped with DEGs detected with RNA-Seq. For IN, we identified 18 genes, including BraA02g005840.3C. For BT, we found thirty genes, including *BraA02g003870.3C* and *BraA02g005190.3C*. For LH, *BraA06g037290.3C* (*GL1*) was identified, similar to the result of Li et al. (2022), who used the extreme pool in the F_2_ population of Chinese cabbage to predict the homologous gene of Arabidopsis *GL1* [16].

## 3. Discussion

In this study, the EMS mutant *lcc*, which had longer hypocotyls and visible internodes in the seedling stage and late budding and late flowering in the reproductive phase, was used to study the related regulatory pathways and genes.

### 3.1. Hormone Genes Related to Internode Elongation

Hormones affect many aspects of seedling development, ranging from seed germination to flowering [6]. CK promotes stem growth by positively regulating cell division [17], and the plant hormone GA represses the DELLA protein in favor of GID1 to advance internode elongation in plants [11,18]. JAZs, repressor proteins of JA signaling, bind with DELLA to affect many growth and development processes in plants [19]. The exogenous application of JAs inhibits various aspects of seedling growth, including primary root growth, leaf expansion, and hypocotyl elongation [19]. Minami et al. (2019) suggested that BRs could boost hypocotyl elongation in *Arabidopsis thaliana* by inducing the phosphorylation of plasma membrane H^+^-ATPase [20]. BRs and AUXs work together in the regulation of plant growth, and there is evidence that BZRs and ARFs can bind together to improve cell elongation [21]. Auxin can cause cells to grow large, and polar auxin transport can be regulated using PINs [22,23].

The results from RNA-Seq revealed that hormone signaling played an important role in the internode growth of Chinese cabbage at the seedling stage.

### 3.2. Link between Budding Time and the Biological Clock

Circadian clocks are endogenous 24 h oscillators that allow organisms to anticipate daily changes in their environment, playing critical roles in many biological processes and stress responses by regulating up to 80% of the transcriptome in plants, which is mainly reflected in growth, flowering, and seasonal rhythm [24]. According to the leaf movement of *lcc* and A03, the circadian rhythm of *lcc* was 1.43 h longer than that of A03 (24 h) [25]. Abnormal expression of *PRRs* impacts flowering time in Arabidopsis, and *prr9* mutants have a longer circadian rhythm [24,26]. Overexpression of *HY5* in Arabidopsis leads to delayed flowering [27]. In Arabidopsis, *CHS* is involved in the regulation of auxin transport [28].

In this study, we found that the KEGG pathways enriched at the early budding and budding stages showed some overlap, which included the circadian clock pathway. The expression of DEGs of the circadian clock pathway was the opposite, i.e., upregulated in T2 and downregulated in T3, which suggested that the circadian clock is involved in budding in Chinese cabbage. Combined analysis with QTLs for BT revealed that *BraA02g003870.3C* and *BraA02g005190.3C* were in QTL regions. *BraA02g003870.3C* (*HY5-1*), which was downregulated in *lcc*, is the homolog of HY5 in Chinese cabbage. *BraA02g005190.3C* (*CHS-1*) is homologous to Arabidopsis *CHS*, was upregulated in the early budding stage, and downregulated in the budding stage of *lcc*. The abnormal expression of *HY5-1* or *CHS-1* may affect flowering and cause later budding in *lcc*.

At present, there are few reports on the internode length and budding time of Chinese cabbage. However, in Chinese cabbage, the length of internodes at the seedling stage and the budding time both affect the quality or formation of leafy heads to varying degrees. Internode elongation is also important for the maintenance of rosette leaves in Chinese cabbage. As a sign, budding time is always complementary with blotting time and flowering time to observe the effects of the cold and photoperiod on reproduction of Chinese cabbage [29,30].

We found that plant hormones and circadian clock pathways directly or indirectly modulate the transcriptional regulation of internode length and budding time. Meanwhile, by combining gene expression and QTL results, candidate genes for these two traits were detected in *lcc*. In fact, there may be other genes that also play a role in the development of *lcc* traits. However, this study can still provide a research basis for the two traits in Chinese cabbage.

## 4. Material and Methods

### 4.1. Plant Material and Growth Conditions for RNA-Seq

The stable mutant *lcc* was derived from A03 using EMS. *Lcc* and wild-type A03 were sown in a field in Hebei, China, on 14 January 2021. During cultivation, these seedlings experienced natural vernalization.

### 4.2. RNA Isolation and Sequencing

Young leaf samples were harvested on the 41st day, 66th day, and 86th day after planting, with three replicates per period and three individuals as one biological replicate. These collected samples were frozen immediately in liquid nitrogen and stored at −80 °C until needed. Total RNA was extracted using an Eastep Super RNA extraction kit (Promega Beijing Biotech, Beijing, China). According to the instructions of EasyScript One-Step gDNA Removal and cDNA Synthesis SuperMix (TransGen Biotech, Beijing, China), reverse transcription was performed to synthesize cDNA. The RNA quality was evaluated using gel electrophoresis and a Nanodrop instrument. Library construction and sequencing were performed by Majorbio.

### 4.3. Bioinformatics Analysis of RNA-Seq Data

The raw sequence data reported in this paper have been deposited in the Genome Sequence Archive (Genomics, Proteomics & Bioinformatics 2021) in National Genomics Data Center (Nucleic Acids Res 2022), China National Center for Bioinformation/Beijing Institute of Genomics, Chinese Academy of Sciences (GSA: CRA015612) which are publicly accessible at https://ngdc.cncb.ac.cn/gsa (accessed on 2 April 2024) [31,32].

Raw data were preprocessed to remove adapters, poly N sequences, and low-quality reads. Clean data for each sample were mapped to the Chinese cabbage reference genome (http://www.bioinformaticslab.cn/EMSmutation/download/ (accessed on 2 May 2022)) using HISAT 2 (2.1.0) [33,34]. The gene expression level was calculated using the fragments per kilobase per million reads (FPKM) method with RSEM (1.3.3) [35]. According to an |log2FC| ≥ 2 and adjusted *p* value < 0.05, a DEG analysis of the two groups (three biological replicates per group) was analyzed using DEGseq2 software (1.24.0), and the results were revised using the BH method.

The software GOATOOLS (0.6.5) was used for a GO enrichment analysis of genes to obtain the main GO functions of the genes. These GO enrichments were checked using Fisher’s test. A KEGG pathway enrichment analysis was performed for genes using R scripts, when the adjusted *p* value was <0.05, GO and KEGG pathways were considered significantly enriched.

### 4.4. RT–qPCR

Total RNA was extracted using an Estep Super (Promega Beijing Biotech, Beijing, China) RNA extraction kit. According to the instructions of EasyScript One-step gDNA Removal and cDNA Synthesis SuperMix (TransGen Biotech, Beijing, China), reverse transcription was performed to synthesize cDNA. To further test the quality of transcription, some genes were selected for qRT–PCR with *ACTIN* as a reference gene. Reactions were performed using the Bio-Rad CFX Connect Optics Module program, and 1 microgram of cDNA template, 0.4 microliters of each primer, 10 microliters of 2× ChamQ Universal SYBR qPCR Master Mix (Novizan Biology, Nanjing, China), and an appropriate amount of sterile water was added to each reaction. Thermal denaturing took place at 95 °C for 30 s, 40 cycles at 95 °C for 10 s and 58 °C for 30 s, and the melting curve was constructed at 95 °C for 15 s, 60 °C for 60 s, and 95 °C for 15 s per cycle, using 2^−ΔΔCt^ to measure the relative gene expression.

### 4.5. Plant Material, Growth Conditions, and Trait Measurements for QTLs

The F_2_ population (n = 150) was developed by crossing the *lcc* mutant of Chinese cabbage A03 (*Brassica rapa* ssp. *pekinensis*) and R500 (*Brassica rapa* ssp. *trilocularis*). The two parental lines and F_2_ individuals were planted under 16 h light 25 °C/8 h dark 18 °C conditions in a greenhouse.

Four traits were investigated in the *lcc*, R500 and F_2_ populations: HL, IN, LH, and BT. The HL of 15 *lcc*, 15 R500, and 150 F_2_ individuals was measured at the seedling stage of cabbage and analyzed with SPSS 23 using a paired samples test (*p* < 0.5) and normal distribution. The IN of F_2_ seedlings depended on the distance between the base of the cotyledon petioles and the base of the second true leaf petioles and was categorized as three types: similar to that of *lcc* or F_1_, longer than that of F_1_ and similar to that of R500. BT was determined by monitoring the first bud in the F_2_ population, and this time it was defined as the first day (two days after the first day, and another cabbage appeared as the first bud; this bud time was described as 3 days). The measurement of FT was the same as that of BT. The details of all traits surveyed are described in Table 2.

### 4.6. DNA Isolation and Resequencing

At the seedling stage, young leaf tissues of the parents and 150 F_2_ individuals were harvested to isolate DNA using the CTAB procedure [36]. DNA quality was detected using gel electrophoresis and a Nanodrop instrument. High-quality DNA was sequenced on the Illumina HiSeq^TM^ PE 150 platform by Novogene Bioinformatics Technology Co., Ltd., Beijing, China, using the TruSeq Library Construction Kit.

### 4.7. Sequencing and Alignment with the Reference Genome

Valid sequencing data were aligned to the *B. rapa* Chiifu reference genome version 1.5 using BWA software (0.7.17) [37]. SAMtools (1.5) [38] was used to change the type of alignment results into SAM/BAM files, and the ratio and coverage were determined a Perl script.

### 4.8. Identification and Annotation of SNPs

To determine if reads from the BWA comparison results had a single location in the Chinese cabbage reference genome, different reads were compared with the Chinese cabbage reference genome. These specific reads were detected using GATK software (3.8) for the population SNP test. To reduce the number of false-positive SNPs caused by sequencing errors, the SNP base support number of parents was set to be no less than 10, and the SNP base support number of offspring was set to be no less than 3. Each SNP detected was annotated using ANNOVAR software (3.0). Based on the results of parental genotyping, polymorphic markers between parents were developed. The loci with missing parental information were filtered, and the loci with homozygous parents and polymorphisms between parents were screened. Genotypes of 150 progenies were extracted from the parental polymorphic marker loci. The presence of abnormal bases in the offspring but not in the parents was regarded as missing data and indicated with the symbol “--”. Screened genotypes covered at least 85% of the markers of all offspring. The candidate markers were filtered using the chi-square test.

### 4.9. Construction of a Genetic Linkage Map and QTL Analysis

Joinmap 4 software (4.1) [39] was used to conduct linkage analysis on the filtered SNP markers to construct a genetic map. Based on the constructed genetic map, QTL analysis was performed on the clock cycle using mapQTL6 software (6) [40]. The LOD threshold for accepting potential QTLs was 3, and the QTL interval was in the range of −1.

## 5. Conclusions

In this paper, we determined the differentially expressed genes between A03 and *lcc*. and conducted QTL using *lcc* and R500. We know that the hormone signaling pathway may be involved in the elongation of internodes and that the circadian clock pathway may regulate the budding of Chinese cabbage. After RNA-Seq and QTL, we found one gene, *BraA02g005840.3C* (*PIN8*), affecting HL and two genes, *BraA02g003870.3C* (*HY5-1*) and *BraA02g005190.3C* (*CHS-1*), affecting BT in Chinese cabbage. However, some mislocation in QTL candidate regions may affect the number and scope of candidate genes, and further research is required to verify these candidate genes. These findings provide a new understanding of internode elongation and budding time in Chinese cabbage.

## Figures and Tables

**Figure 1 plants-13-01083-f001:**
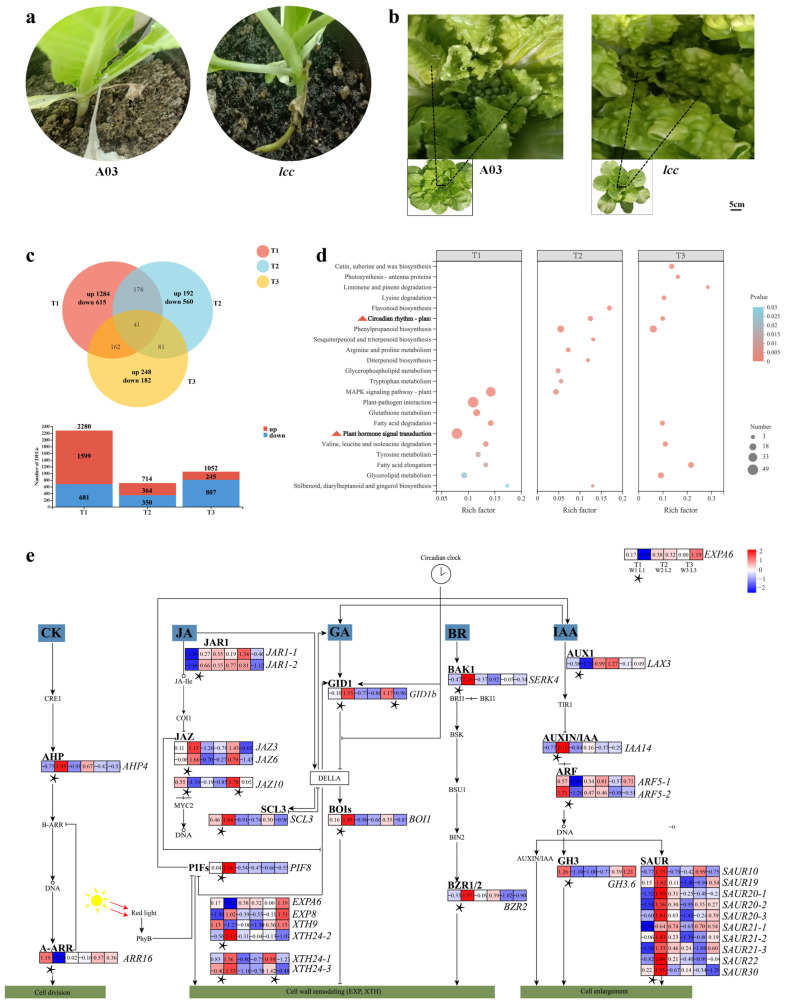
Related traits and RNA-Seq information. (**a**,**b**) Morphological characteristics of Chinese cabbage A03 and the *lcc* mutant. (**a**) Internodes of A03 and *lcc*. (**b**) Early budding time of A03 and *lcc*. (**c**) Venn diagram of DEGs and up- and downregulated genes in three stages. (**d**) KEGG analysis. (**e**) Signaling pathway related to internode growth. Asterisk indicated the stage of DEGs at *p* < 0.05. Full line indicated molecular interaction or relation.

**Figure 2 plants-13-01083-f002:**
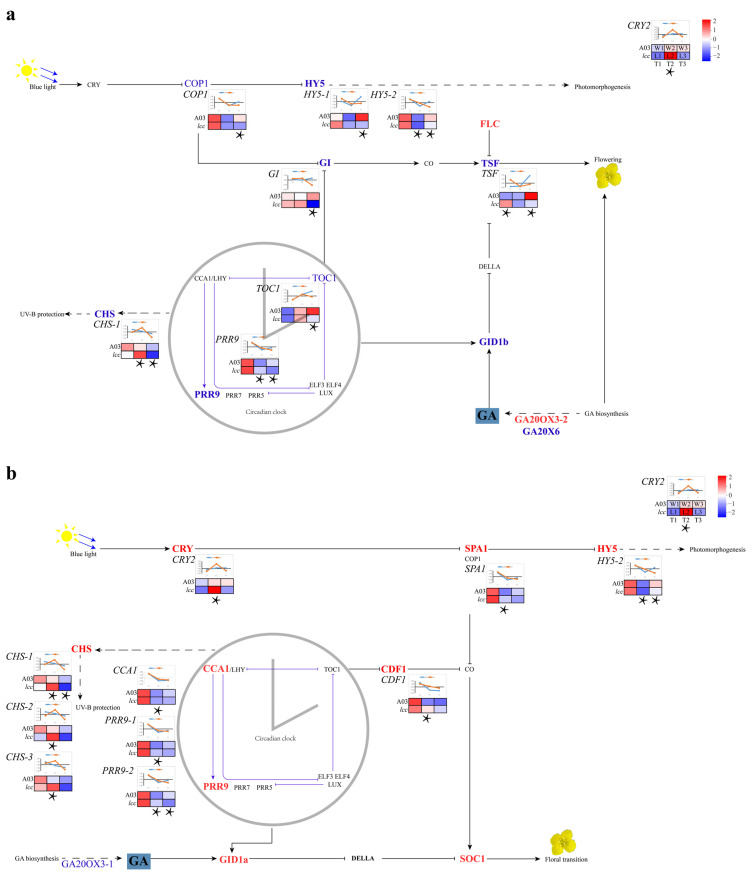
Signaling related to flowering. Up- and downregulated genes were marked in red and blue, respectively. The expression of DEGs related to circadian clock was supplemented with the graph. (**a**) Pathway related to flowering in the early budding stage. (**b**) Pathway associated with flowering in the budding stage. Asterisk indicated the stage of DEGs at *p* < 0.05. Full line indicated molecular interaction or relation, dotted line indicated indirect link or unknown relation.

**Figure 3 plants-13-01083-f003:**
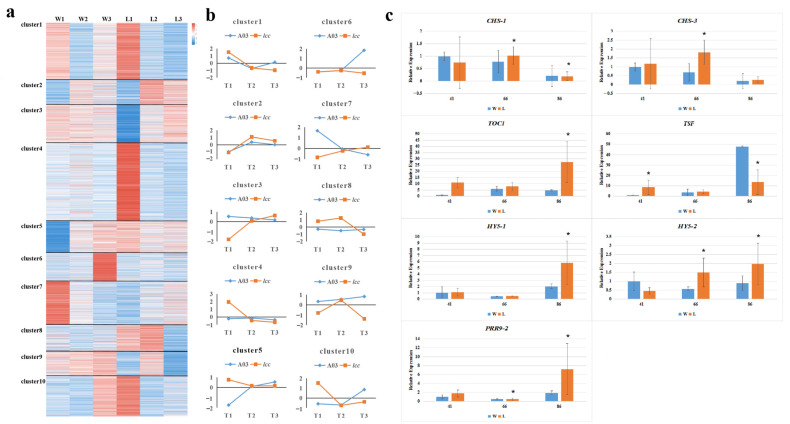
Cluster analysis and validation. (**a**) Cluster analysis of DEGs between A03 and *lcc*. (**b**) Gene expression patterns for ten clusters. (**c**) RT–qPCR validation of eight DEGs. Asterisk indicated the difference is significant at *p* < 0.05.

**Figure 4 plants-13-01083-f004:**
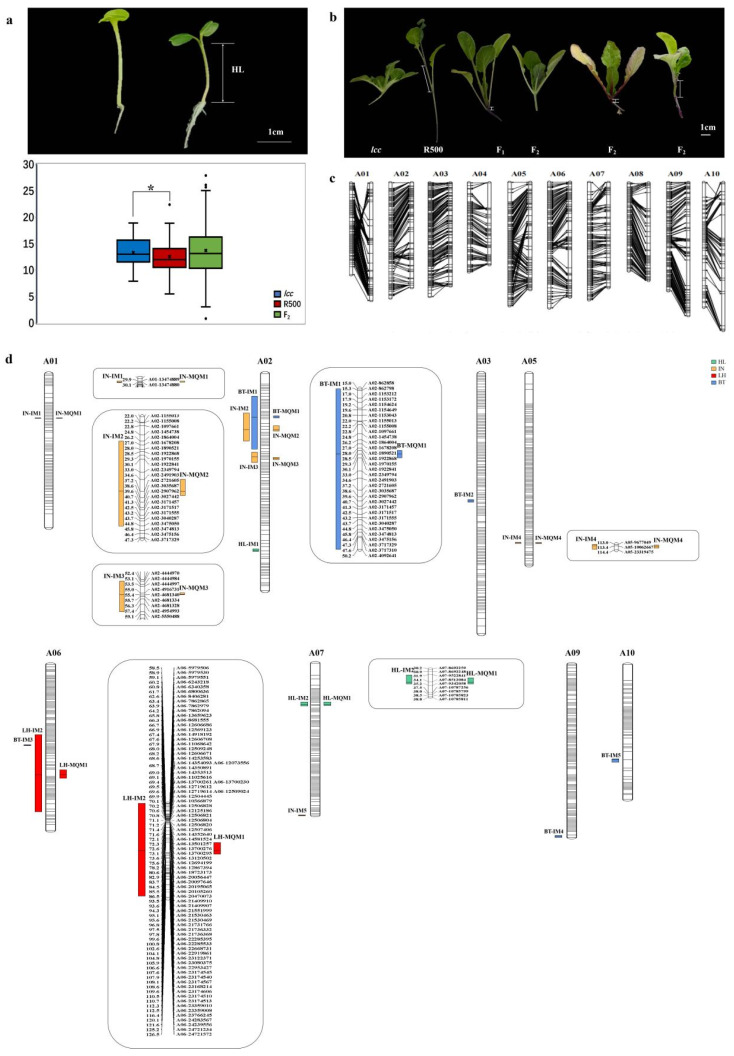
Analysis of related traits and QTLs for four traits discovered in the F_2_ population. (**a**) Hypocotyl length data for five-day-old seedlings of *lcc*, R500, and F_2_. Asterisk indicated the internode length difference between *lcc* and R500 is significant at *p* < 0.05. (**b**) Internode variation of *lcc*, R500, F_1,_ and F_2_. The internode length of F_2_ individuals was between those of *lcc* and R500. (**c**) The order of SNP markers in the linkage map (**left**) and physical map (**right**). (**d**) The QTLs for HL, IN, and BT on the linkage map. IM QTLs are on the left, and MQM QTLs are on the right.

**Table 1 plants-13-01083-t001:** Primers and genes for RT–qPCR.

Gene Name	Gene ID	Forward Primer	Reverse Primer
Actin	BraA02g003190.3C	GGAGCTGAGAGATTCCGTTG	GAACCACCACTGAGGACGAT
CHS-3	BraA10g024990.3C	CGCGTGTGTTCTCTTCATATTGG	CAAGACCACTGTCTCTACGGTAAG
CHS-1	BraA02g005190.3C	GAGGAAGTCTAAGGAAGATGGTGTG	TTAGACAGGAACGCTGTGTAGG
PRR9-2	BraA05g001070.3C	GAGAAGCAAGATCAAACCACCAAG	GCTGCCTGGCTGTTCTCATA
HY5-1	BraA02g003870.3C	AAGAGACCAAGCGGCTAAAGAG	CTCTAAGTCTTTCACTCTGGTCTCC
HY5-2	BraA05g029990.3C	CGAGGGAGAGGAAGAAAGTGTATG	TTAGTGATTGTCGTCAGCTTTAGGC
TOC1	BraA03g044360.3C	GTCATGTGCCTTTACAGAATGGTC	GCTTAGTCACTCTCACCTCGTT
TSF	BraA07g031650.3C	AACCCGCACCTTCGAGAATATC	GAACAATACCAGCACGAGACGA

**Table 2 plants-13-01083-t002:** Summary of traits and measurements.

Trait Name	Abbreviation	Measurement
Hypocotyl length	HL	Length of the hypocotyl (the part of a seedling below the cotyledon, above the root), measured using a ruler at the seedling stage (cm).
Internode length	IN	The distance between the base of the cotyledon petioles to the base of second true leaf petioles, scored as 1, looks like that of *lcc* or F_1_; 2, longer than that of F_1_; or 3, looks like that of R500. Measured by eye at the seedling stage.
Leaf hairs	LH	Determined with visual assessment.
Budding time	BT	Days after the first bud of the F_2_ population appeared (days).

## Data Availability

The data presented in this study are available on request from the corresponding author.

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
