# Peer review of "Transcriptional Regulation and Gene Mapping of Internode Elongation and Late Budding in the Chinese Cabbage Mutant *lcc"

_plants, 2024, doi:10.3390/plants13081083_

Round 1
Reviewer 1 Report
Comments and Suggestions for Authors
Dear Authors,
the manuscript “Transcriptional Regulation and Gene Mapping of Internode Elongation and Late Budding in the Chinese cabbage mutant lcc” is well written and reports interesting information relating to the transcriptional regulation of internode length and budding time in Chinese cabbage.
In my opinion, the manuscript could be accepted for publication after minor revisions to the text and figures.
First of all, it would be appropriate to indicate in which database the raw reads produced by RNA-Seq are archived.
Line 3 Please, change "abbage" to "cabbage"
Line 21 Please, explains the acronym EMS as (Ethyl Methane Sulfonate)
Line 58 Write in italics “osarp6”
Figure 1 The figures are too small and difficult to read
Line 160 Write in italics “lcc”
Line 161 Write in italics “lcc”
Figure 2 Please, add GI gene graph
Line 164 Write in italics “lcc”
Line 166 Please, explains the acronym BT as (Budding Time)
Line 227 Please, change “Li and Wang” to “Li et al.”
Line 247 Please, change “Minami A and Takahashi K” to “Minami et al.”
Line 248 Write in italics “Arabidopsis thaliana”
Line 264 Write in italics “PRR” and “prr9”
Line 288 Add “4.” to Material and Methods
Line 333 Please, write in italics “Brassica rapa” and “pekinensis”
Line 429 Remove “2004”
Line 433 Remove “2019”
Line 441 Remove “2006”
Line 447 Remove “2006”
Line 465 Remove “2020”
Reviewer 2 Report
Comments and Suggestions for Authors
The manuscript “Transcriptional Regulation and Gene Mapping of Internode Elongation and Late Budding in the Chinese abbage mutant lcc” is dealing with comparison of gene expression patterns of mutant (lcc) with wild-type (A03). Authors identified 2280 differentially expressed genes were identified in the seedling stage, 714 differentially expressed genes were identified in the early budding stage, and 1052 differentially expressed genes were identified in the budding stage.
It was found that BraA02g005840.3C gene, a homologous gene of PIN8 in Arabidopsis, related to auxin transporter is downregulated in lcc at the seedling stage. This gene is assigned as potential regulator of internode length in Chinese cabbage. In addition, BraA02g003870.3C (HY5-1) and BraA02g005190.3C (CHS-1) were identified as budding time candidate genes. After RNA-Seq and QTL, we found one gene, BraA02g005840.3C (PIN8), affecting length of the hypocotyl and two genes, BraA02g003870.3C (HY5-1) and BraA02g005190.3C (CHS-1), affecting first bud appearance in Chinese cabbage.
The manuscript belongs to the scope of the journal Plants. The findings has some merit for understanding of internode elongation and budding time in Chinese cabbage. The manuscript can be accepted for publication after major revision.
Here are some suggestions for manuscript improvement which are highlighted in text:
Line 3: In tittle there is one letter in word abbage?
Line 69: There is something wrong with “, let alone…
Figure 1: These Figure is too big. I suggested to divide in two: a, b to be separate Figure which can be optionally incorporated in Material and methods section, 4.1
Line 181: Start sentence with uppercase letter
Line 278: Please, incorporate and appropriately discus all relevant reports about internode length and budding time in Chinese cabbage. It will be very useful for readers
Line 290: Start sentence with uppercase letter
Line 353: B. rapa, italic

Round 2
Reviewer 2 Report
Comments and Suggestions for Authors
Authors significantly improved first version of manuscript.